# Single-cell sequencing reveals clonal expansions of pro-inflammatory synovial CD8 T cells expressing tissue-homing receptors in psoriatic arthritis

Frank Penkava [1], Martin Del Castillo Velasco-Herrera [2], Matthew D. Young[2], Nicole Yager[1], Lilian N. Nwosu [3], Arthur G. Pratt[3,4], Alicia Lledo Lara[5], Charlotte Guzzo [2], Ash Maroof[6], Lira Mamanova[2], Suzanne Cole[6], Mirjana Efremova[2], Davide Simone[1], Andrew Filer [7], Chrysothemis C. Brown[8], Andrew L. Croxford[9], John D. Isaacs[3,4], Sarah Teichmann [2], Paul Bowness [1,12], Sam Behjati[2,10,11,12✉] & M. Hussein Al-Mossawi [1,12✉]

Psoriatic arthritis (PsA) is a debilitating immune-mediated inflammatory arthritis of unknown pathogenesis commonly affecting patients with skin psoriasis. Here we use complementary single-cell approaches to study leukocytes from PsA joints. Mass cytometry demonstrates a 3-fold expansion of memory CD8 T cells in the joints of PsA patients compared to peripheral blood. Meanwhile, droplet-based and plate-based single-cell RNA sequencing of paired T cell receptor alpha and beta chain sequences show pronounced CD8 T cell clonal expansions within the joints. Transcriptome analyses find these expanded synovial CD8 T cells to express cycling, activation, tissue-homing and tissue residency markers. T cell receptor sequence comparison between patients identifies clonal convergence. Finally, chemokine receptor CXCR3 is upregulated in the expanded synovial CD8 T cells, while two CXCR3 ligands, CXCL9 and CXCL10, are elevated in PsA synovial fluid. Our data thus provide a quantitative molecular insight into the cellular immune landscape of psoriatic arthritis.

[1] Nuffield Department of Orthopaedics Rheumatology and Musculoskeletal Sciences, University of Oxford, Oxford OX3 7LD, UK. [2] Wellcome Sanger Institute, Hinxton CB10 1SA, UK. [3] Translational and Clinical Research Institute, Newcastle University, Newcastle upon Tyne NE2 4HH, UK. [4] Musculoskeletal Unit, Newcastle upon Tyne Hospitals NHS Foundation Trust, Newcastle upon Tyne NE7 7DN, UK. [5] Wellcome Centre for Human Genetics, University of Oxford, Oxford OX3 7BN, UK. [6] UCB Pharma, 216 Bath road, Slough SL1 3WE, UK. [7] NIHR Birmingham Biomedical Research Centre, University Hospitals Birmingham NHS Foundation Trust and University of Birmingham, Institute of Inflammation and Ageing, Birmingham, UK. [8] Infection, Inflammation and Rheumatology Section, UCL Great Ormond Street Institute of Child Health, London WC1N 1EH, UK. [9] Idorsia Pharmaceuticals Ltd., Drug Discovery Immunology, Hegenheimermattweg 91, 4123 Allschwil, Switzerland. [10] Cambridge University Hospitals NHS Foundation Trust, Cambridge CB2 0QQ, UK. [11] Department of Paediatrics, University of Cambridge, Cambridge CB2 0SP, UK. [12] These authors contributed equally: Paul Bowness, Sam Behjati, M. Hussein Al-Mossawi. ✉email: sb31@sanger.ac.uk; Hussein.al-mossawi@ndorms.ox.ac.uk

Up to one-third of patients suffering from psoriasis develop the debilitating immune-mediated inflammatory joint disease, psoriatic arthritis (PsA)[1]. The pathogenesis of PsA is complex, involving multiple inflammatory pathways, and is thought to be driven by immune cells entering or proliferating within the joint synovial lining tissue and/or fluid[2]. Genome-wide association studies of both psoriasis and PsA support a pathogenic role for CD8 T cells, showing significant associations with MHC class I allotypes and with other genes involved in T-cell function[3]. Furthermore, clonal expansions of CD8 cells have been reported in the synovial fluid of patients with PsA using bulk T-cell receptor beta-chain sequencing approaches[4]. Costello and colleagues identified an average of 32 private T-cell receptor (TCR) beta-chain clonal expansions unique to individual patients and primarily confined to synovial fluid. While they found evidence supporting antigen-driven selection, with differing nucleic acid sequences encoding common amino acid sequences, they did not see evidence of shared or public repertoires between different patients, as previously described for HLA-B27-associated spondyloarthritis[5]. Leucocytes are recruited to the synovium and synovial fluid through chemokine gradients, with bulk synovial fluid cell pellet RNA showing increased expression of the CXCR3 gene and synovial fluid showing increased expression of one of its protein ligands CCL10[6]. A subgroup of patients with PsA develop a distinct pattern of arthritis affecting predominantly large joints termed large-joint oligo PsA[7]. These patients often require therapeutic knee-joint aspiration that provides an opportunity to examine the synovial fluid exudate and tissue in PsA.

Here, we study the cellular landscape of PsA blood, synovial fluid and synovial tissue at single-cell resolution, combining mass cytometry with chromium 10× genomics (10×) droplet-encapsulated single-cell mRNA sequencing and validation by full-length transcript Smart-seq 2 (SS2) single-cell mRNA sequencing. We use 5′ chromium technology to unravel the alpha-/beta-paired chain clonal architecture of T cells in the synovial fluid compared to peripheral blood and match this with gene expression. We identify clonal expansions of CD8 T cells in PsA synovial fluid, show that single alpha–beta T-cell sister clones exist in multiple cell states and highlight pathways that mediate their trafficking into the joint. These findings have implications for our understanding of the pathogenesis of PsA and other HLA-associated immune-mediated inflammatory diseases, as well as suggesting potential therapeutic approaches.

## Results

**Cellular landscape of PsA.** Freshly isolated peripheral blood and synovial fluid were obtained from patients with large-joint oligo PsA presenting for arthrocentesis, and immediately (0.5–4 h) processed for mass cytometry (CyTOF) and single-cell RNA sequencing (Fig. 1a, b). We used CyTOF to quantify leucocyte populations from matched synovial fluid and blood obtained from ten patients. Blood and synovial fluid samples were fixed within 30 min of collection, stained with a 38-marker antibody panel (Supplementary Table 2) and then acquired on a CyTOF Helios instrument. After pre-processing (Supplementary Fig. 1a), we used t-stochastic neighbour embedding (t-SNE) to derive cell clusters (Fig. 1c), which were annotated by FlowSOM[8] (Supplementary Fig. 1b, c). These analyses identified significant expansions of synovial memory CD8 ($p = 0.0059$, paired t test) and memory CD4 ($p = 0.025$, paired t test) T cells (Fig. 1d, e) in all patients compared to blood. Plasmacytoid ($p = 0.032$, paired t test) and conventional dendritic cells ($p = 0.013$, paired t test) were also expanded in synovial fluid. B cells and basophils were depleted ($p = 0.0059$ for both, paired t test), and monocytes, gamma–delta T, mucosal invariant T (MAIT)[9] and NK cells were

unchanged (Fig. 1d). 3′ droplet-encapsulated single-cell mRNA sequencing of PBMC and SFMC from three PsA patients, carried out in parallel, confirmed the presence of these cell types and did not identify any additional cellular populations (Supplementary Fig. 2a–c, Supplementary Data 1a).

**Sequencing of PsA blood, synovial fluid and tissue T cells.** Supported by the genetic association of PsA with polymorphisms in T-cell-related genes[10] and the significant expansions observed in our CyTOF analysis (Fig. 1d), we specifically interrogated the transcriptional profile of synovial fluid memory CD4 and CD8 T-cell populations. For three patients, we used droplet-encapsulated single-cell 5′ mRNA sequencing (chromium 10×), with Smart-seq 2 validation in four patients (applying both 10× and Smart-seq 2 technology in parallel on the same sorted cells for one donor). For both approaches, synovial fluid and blood were processed in parallel directly ex vivo within 4 h, with single-cell suspensions enriched for CD4 and CD8 T cells by flow cytometry-activated cell sorting (FACS, Supplementary Fig. 3a). In addition, we analysed CD4 and CD8 T-cell populations identified within CD45+ sorted cells from the cryopreserved PsA knee synovial tissue biopsies of two further patients, also using 5′ chromium 10× technology (Supplementary Fig. 3b, c, Supplementary Fig. 4, Supplementary Data 1b). After applying stringent quality-control criteria ("Methods"), we performed a unified analysis of 41,202 single T-cell transcriptomes of equal patient origin from the paired blood and synovial fluid samples, together with 251 T-cell transcriptomes from the synovial tissue biopsies (clusters 2, 3 and 8 from Supplementary Fig. 4a also expressing CD3E transcripts) using the Seurat 3 pipeline. We identified 16 clusters of memory CD4 and CD8 T cells (Fig. 2a), annotated with key marker genes in Fig. 2b (Supplementary Fig. 5, Supplementary Data 1c and 2). Of note, one cluster (cluster 16), primarily composed of synovial CD8 T cells, was distinguished by high expression of the proliferation markers MKI67 and STMN1[11], indicating active proliferation of CD8 T cells within inflamed joints. We observed accurate alignment of peripheral blood and synovial fluid T cells, with cells originating from both sample types present in all clusters for all patients (Fig. 2c, d, Supplementary Fig. 6). Synovial tissue T cells also mapped to the majority of clusters identified, although synovial tissue CD8 T cells predominantly mapped to cluster 2 (HLA-DR-low CD8) (Fig. 2e, Supplementary Data 2). Comparing gene expression between synovial fluid and blood T cells, we observed increased expression of activation and effector markers in synovial fluid in the HLA-DR-low CD8, HLA-DR-high CD8 and ZNF683+ CD8 clusters (Fig. 2f–h, Supplementary Data 3a–c). In particular, we observed significantly upregulated expression of the MHC-II genes HLA-DRB1 and HLA-DRA in the synovial HLA-DR-high CD8 cluster, and increased expression of the effector molecules GZMA and GZMB in the synovial fluid ZNF683+ CD8 cluster. Of note, the T-cell receptor alpha-chain gene TRAV27 was significantly upregulated in the synovial fluid compartment of the ZNF683+ CD8 cluster compared to peripheral blood, suggestive of synovial clonal expansion within this cluster.

**CD8 T-cell clonal expansion in PsA synovial fluid.** We used T-cell receptor VDJ sequencing to understand the clonal composition of PsA blood and paired synovial fluid (Fig. 3a, b), mapping paired alpha-/beta-chain TCR sequences to gene expression for the same cells. We first looked systematically for evidence of clonal expansion of CD4 and CD8 T cells in the blood and synovial fluid for each of the three donors using the 10× data set (Fig. 3c–f). For every individual, we observed between 7 and 20 CD8 clones (and 1–4 CD4 clones) significantly enriched in the

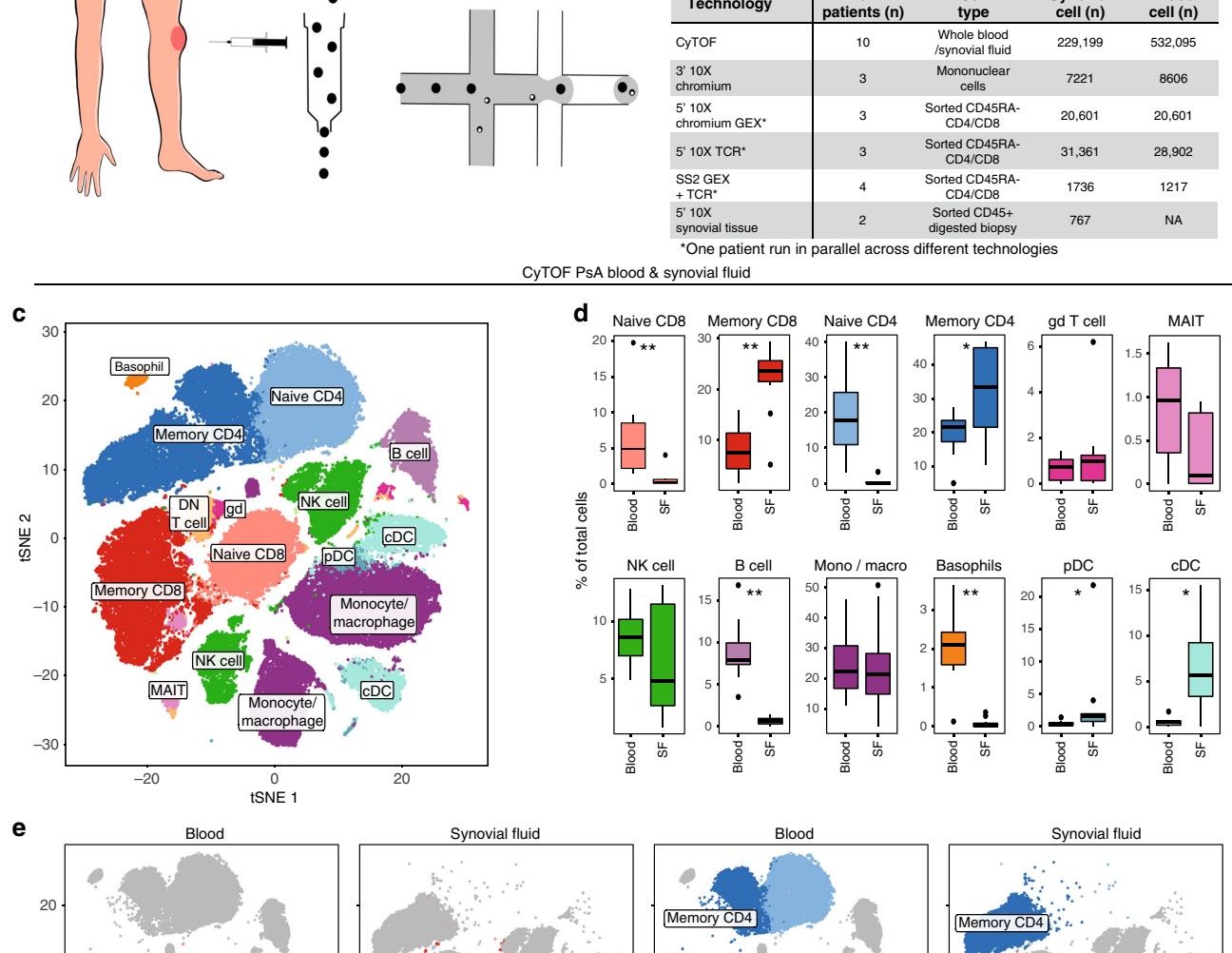

**Fig. 1 Landscape of synovial leukocyte populations in psoriatic arthritis. a** Overview of experimental design. **b** Cell numbers used in each of the experimental techniques**. c** Representative map of CyTOF clusters derived from one PsA patient's matched peripheral blood and synovial fluid cells using *t*-SNE. **d** Boxplots showing cluster frequencies within blood and synovial fluid (SF) from ten independent experiments. All replication attempts were successful. For all boxplots, the upper whisker extends to the largest value, maximally 1.5 × IQR from the 75th centile, and the lower whisker extends to the smallest value, 1.5× IQR from the 25th centile. The junctions of whisker and box (hinge) represent the 25th and 75th centiles, while median is indicated by the central line. Outlier values are plotted individually beyond the whiskers. Two-sided paired *t* test with Bonferroni correction. $n = 10$ (* = $p < 0.05$, **=$p < 0.01$, ***=$p < 0.001$). Exact $p$ value naive CD8, memory CD8, naive CD4, B cells and basophils = 0.0059, memory CD4 = 0.025, pDC = 0.032 and cDC = 0.013. **e** Representative map of CyTOF clusters derived from one PsA patient, divided according to tissue of origin and highlighting memory CD8 (dark red) and memory CD4 (dark blue) T cells. Source data are provided as a Source Data file.

synovial fluid; we also observed a lesser number of clones enriched in the peripheral blood (see "Methods", Supplementary Data 4 and 5). Overall, cells from synovial-enriched clones were most likely to be part of the HLA-DR-high, CD8 cycling or ZNF683+ CD8 clusters (odds ratios 12.6, 4.73 and 3.66; adjusted *p* value 0, 8.7E−19 and 6.1E−64, respectively, two-sided Fisher's exact test with Bonferroni correction, Supplementary Data 7a, Fig. 3c). Looking at individual clones, cells from the majority of synovial fluid-enriched clones were present in multiple phenotypic clusters (Fig. 3g). This was also the case for clones enriched

in peripheral blood. However, the HLA-DR-high and cycling clusters were more often overrepresented by synovial-enriched clones, while HLA-DR-low and transitional cluster phenotypes were more common in peripheral blood-enriched clones. A smaller number of enriched clones represented cells from the ZNF683+ and central memory clusters for both sample types (Supplementary Fig. 7). In addition, we carried out VDJ receptor sequencing of CD4 and CD8 cells from the synovial tissue. Due to the small number of cells obtained by biopsy, the largest clone observed came from one sample and mapped to four CD8 cells

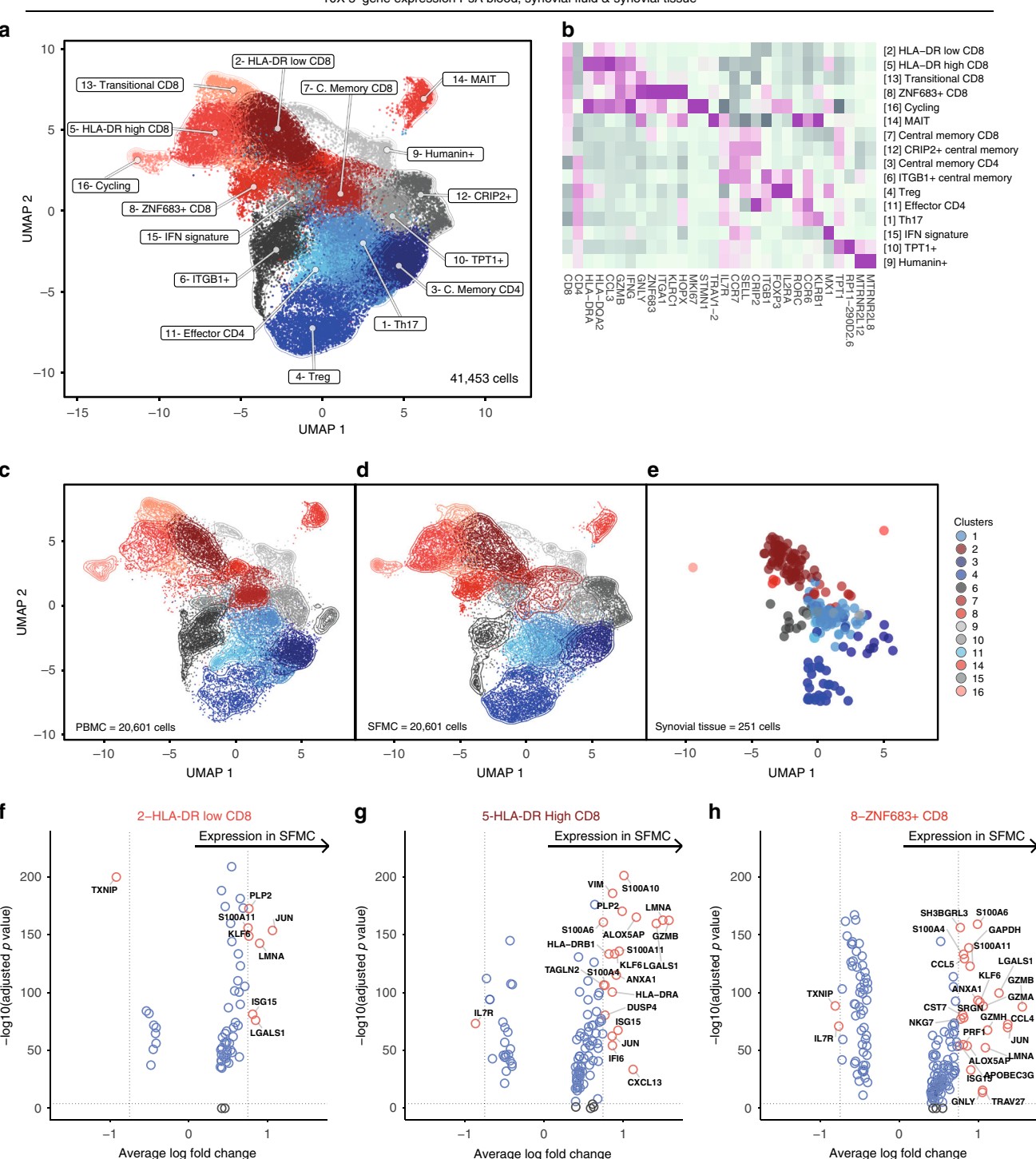

**Fig. 2 Transcriptional landscape of CD4 and CD8 T cells in psoriatic arthritis. a** UMAP of integrated PsA blood, synovial fluid and synovial tissue memory CD4 and CD8 T cells (paired blood and synovial fluid from three patients; synovial tissue from two other patients). Clusters coloured red comprise CD8 cells, clusters coloured blue are CD4 cells. Clusters in grey contain mixed CD4 and CD8 populations. **b** Heatmap showing memory CD4 and CD8 immune subset signatures. The relative expression of marker genes (columns) across cell clusters (rows) is shown. **c**–**e** UMAPs showing the contribution of PBMC, SFMC and synovial fluid-derived T cells to the integrated UMAP from (**a**). **f**–**h** Volcano plots showing differential expression of genes between blood and synovial fluid T cells in HLA-DR-low (**f**), HLA-DR-high (**g**) and ZNF683+ (**h**) clusters (Supplementary Data 3a–c). Statistics calculated using two-sided Wilcoxon Rank Sum test with Bonferroni correction. Genes where adjusted p value < = 1E−04 and average logFC is greater than 0.75 or less than −0.75 are labelled and shown as red circles. Dotted lines indicate significance and average logFC cut-off for displaying labelled genes (paired blood and synovial fluid from three patients). Source data are provided as a Source Data file.

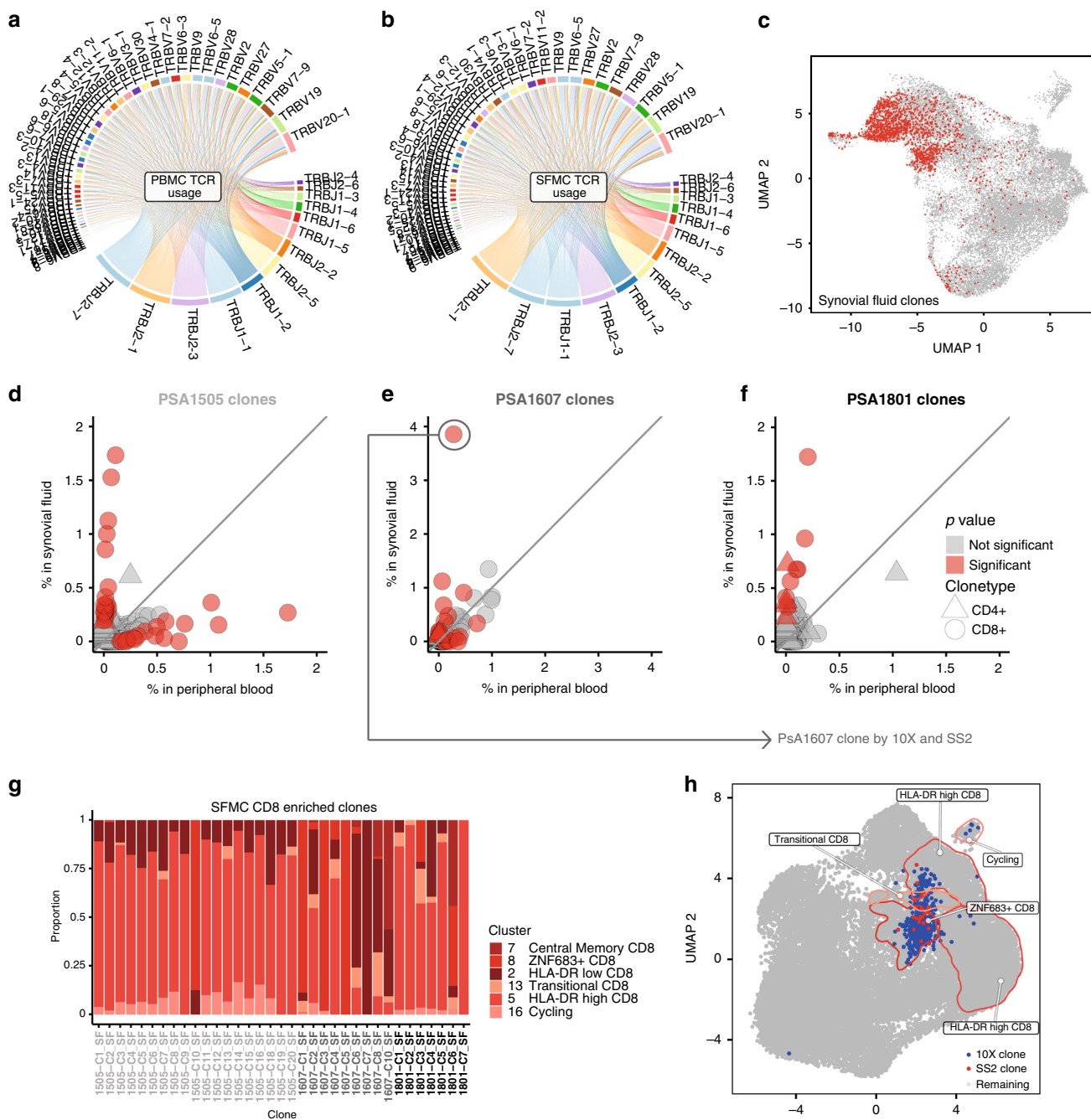

**Fig. 3 Clonal expansion of CD8 T cells in psoriatic arthritis. a**, **b** TCR beta-chain V and J-gene usage in PsA blood and synovial fluid generated from 10× 5′ data of patients PSA1505, PSA1607 and PSA1801. V and J segments are represented by arcs, scaled in size to their combined usage by all patients within each sample type, and giving equal weight to each patient. V–J pairings are represented by ribbons. Plot generated using VDJtools (v1.2.1)[36]. **c** UMAP of PsA synovial fluid CD4 and CD8 T cells from three patients with clonally expanded cells highlighted in red. Clonally expanded cells in synovial fluid associated most strongly with HLA-DR-high CD8, cycling and ZNF683+ CD8 clusters (odds ratios 12.6, 4.73 and 3.66; adjusted $p$ value 0, 8.7E−19 and 6.1E−64, respectively; two-sided Fisher's exact test with Bonferroni correction using R stats package; Supplementary Data 7a). **d**–**f** Individual clonal expansion across blood and synovial fluid for three PsA patients based on 10× 5′ data. Circles represent CD8 clonotypes, triangles represent CD4 clonotypes. Data points coloured red show significantly expanded clonotypes (adjusted $p$ value ≤ 0.05). Two-sided Fisher's exact test, Benjamini–Hochberg correction. **g** Cluster assignment of enriched CD8 clones in PsA synovial fluid from three patients (PSA1505, PSA1607 and PSA1801). **h** UMAP plot of integrated memory T cells from one donor (PSA1607), including cells from 5′ 10× and SS2 datasets (Supplementary Fig. 8). Synovial CD8 T cells from the clone most enriched in synovial fluid for this patient are highlighted in blue for the 10× data set and red for the SS2 data set. Source data are provided as a Source Data file.

with a HLA-DR-low phenotype and one cell with a cycling phenotype (Supplementary Data 5f). For patient PSA1607, we combined 10× and Smart-seq 2 gene-expression data to compare gene expression and clonality across platforms (Supplementary Fig. 8a–d). We observed cells from the clone maximally enriched in synovial fluid from each technology within the same clusters and with a similar magnitude of clonal expansion, validating our approach (Fig. 3h, Supplementary Fig. 8e, Supplementary Data 4e, g).

**T-cell receptor convergence in PsA synovial clones**. To determine whether clonal expansion of CD8 or CD4 T cells may be driven by common antigen(s), we used the grouping of lymphocyte interactions by paratope hotspots (GLIPH)[12] algorithm to assess TCR complementarity-determining region 3 (CDR3) similarity and putative shared specificity across all T cells studied in the 10× experiment (Supplementary Data 6). GLIPH analysis of 40915 synovial cells with 19582 unique CDR3 beta-chain amino acid sequences revealed 143 TCR convergence groups (CRG) with shared specificity between the three patients (Supplementary Data 6a–d). One CRG in particular contained a high number of cells belonging to synovial-enriched CD8 clones, including the most enriched clones from two patients ($p = 4.2E$ $-27$ and $2.1E-112$ for patients PSA1505 and PSA1607, respectively). GLIPH also identified a CDR3 (NQNT) motif in this CRG (observed vs. expected fold change 10.869, $p = 0.001$ relative to the expected frequency in an unselected naive reference TCR set[12]). To support these specificity groups, we studied a further 1441 synovial CD4 and CD8 T cells with 1236 unique beta-chain CDR3 amino acid sequences from three independent patients in the Smart-seq 2 data set (Supplementary Data 6e, f). GLIPH analysis incorporating these sequences with the original 10× TCR sequences obtained from 40,915 cells identified 5 TCR specificity groups common to all 6 patients (Fig. 4a and Supplementary Data 6g–j shows the distribution and make-up of the CRGs across the 6 patients in the two datasets. One of these five groups (CRG-1) was again assigned the NQNT motif and incorporated the same clones as the synovial-enriched CRG identified by droplet-encapsulated data alone (Supplementary Data 6k). CRG-1 contributed the greatest number of expanded clones to the observed CD8 T-cell expansions in synovial fluid and displayed high usage of the TCR genes *TRBV28* and *TRBJ1–1* (Supplementary data 6g–j, Fig. 4b). Of note, cells from CRG-1 were predominantly and disproportionately assigned to the ZNF683+ and HLA-DR- high synovial cell clusters (odds ratios 4.5 and 1.7, respectively; adjusted $p$ value $3.2E-57$ and $4.2E-07$, respectively, Fisher's exact test with Bonferroni correction (Fig. 4c, d, Supplementary Data 7b)). The HLA-DR-high cluster 5 was defined by transcripts indicating an activated phenotype, including granzymes (*GZMA*, *GZMB*, *GZMH* and *GZMK*), *CCL4*, *CCL5*, *CD74* and MHC-II (Fig. 4e, Supplementary Data 3d). ZNF683+ cluster-defining transcripts (Fig. 4f, Supplementary Data 3e) included *KLRC1* (NKG2A), the tissue-residency marker *ZNF683*[13], the skin/gut-homing marker *ITGA1* (CD49a)[14] and granulysin (*GNLY*). The cells within CRG-1 showed a similar gene- expression profile compared to their non-CRG-1 neighbours within the same cluster (Supplementary Fig. 9a). Cells from both the HLA-DR-high and ZNF683+ clusters showed a gene signature overlapping with previously described tissue-resident epidermal skin CD49a+ CD8 T cells poised for cytotoxic function, which were enriched for specific TCR V-gene usage including *TRBV27* and *TRBV28*[15] (Supplementary Fig. 9b).

**Chemokine receptor expression and quantification**. Comparison of blood and synovial fluid TCR clonotypes showed that some clonotypes were predominantly enriched in blood, while others were predominantly enriched in the synovial fluid. To look for mRNAs that might mediate trafficking, we compared transcriptomes of synovial and blood T cells from clones significantly enriched in synovial fluid ($n = 1766$ cells) and blood ($n = 842$ cells), respectively. Subsetting and reclustering of these 2608 cells yielded 8 clusters (Fig. 5a, b, Supplementary Data 1e) mapping back to the original clustering analysis (Fig. 2a). *CXCR3* was the most strongly expressed chemokine receptor gene in synovial-enriched T-cell clones (avg. logFC = 0.63, adjusted $p = 1E-58$, two-sided Wilcoxon rank-sum test with Bonferroni correction), followed by *CXCR6*, *CCR5* and *CCR2*, while *CX3CR1* (avg. logFC = 0.66, adjusted $p = 4.7E-62$) and *CXCR5* were more strongly expressed in peripheral blood-enriched clones (Supplementary Data 1f). Intercluster variation of chemokine receptor gene expression within these compartments is visualised in Fig. 5c. The overexpression of *CXCR3* in synovial T cells was striking (avg. logFC = 0.7, adjusted $p = 0$, two-sided Wilcoxon rank-sum test with Bonferroni correction) when mapped back to the whole 10× 5′ data set (Fig. 5d, e, Supplementary Data 1g), and was also present in synovial cells of the dominant CRG-1 GLIPH CRG (Supplementary Fig. 9a). To further expand on this finding, we measured protein levels of IP10 (CXCL10) and CXCL9 (both ligands for CXCR3), together with MIP1α (CCL3, a ligand for CCR1, CCR4 and CCR5) and MIP1β (CCL4, a ligand for CCR5), in the plasma and synovial fluid of patients with PsA (Fig. 5f). Both CXCR3 ligands were highly enriched in the synovial fluid compared to blood ($p = 0.0004$ for CXCL10, $p = 0.007$ for CXCL9, paired $t$ test).

## Discussion

In this study, we observe striking expansions of memory CD8 and CD4 T cells in the synovial fluid of patients with large-joint PsA. The parallel use of complementary single-cell approaches decreased the chances of bias, in particular through the use of unsupervised clustering approaches for the CyTOF data, and followed by validation in a smaller subset of patients using different single-cell mRNA sequencing techniques.

The large number of T cells sequenced from the synovial fluid and blood of patients with PsA allowed detailed insight into the cell states found in the joint. Intriguingly, we observe cells in the synovial fluid, synovial tissue and the peripheral blood that fall into clusters that have markers of tissue residence and activation. The existence of these memory phenotypes in the blood is not surprising, given the exposure of adult human to chronic viral infections such as EBV and CMV[16]. When comparing cells falling within these clusters with their synovial fluid counterparts, we observe the synovial fluid cells to have increased expression of cytokines and other markers of effector function including granzymes.

The use of T-cell receptor VDJ sequencing in this study allows us to understand the clonal architecture of PsA with a level of detail not previously possible. We replicate and extend observations of clonal expansions of CD8 T cells in the joints of patients with PsA, previously suggested using bulk sequencing techniques[4], and now have the ability to track single alpha–beta clones across the blood and the synovial fluid. Whilst we observe some hints of clonal expansion in the synovial tissue, the cell numbers obtained from synovial biopsy in this study prevented us from studying synovial tissue clonality. Within the synovial fluid, we here show that almost all expanded CD8 T-cell clones exist in multiple cell states. Our findings are consistent mechanistically with previous observations of asymmetric T-cell division[17]. The presence of expanded clones expressing markers of activation, tissue residency and/or tissue homing with evidence of

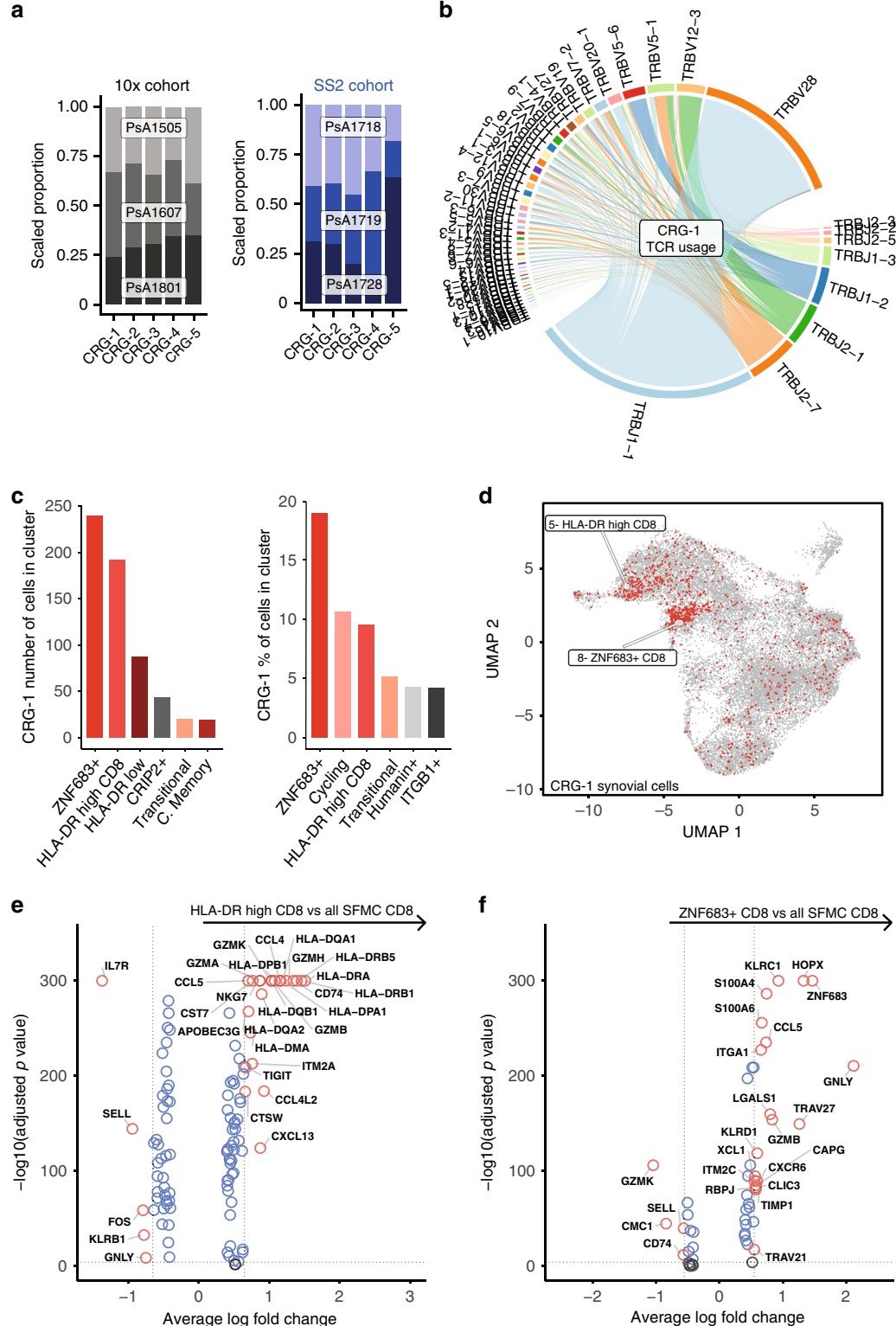

**Fig. 4 T-cell receptor convergence in PsA synovial clones. a** Patient distribution of top 5 GLIPH convergence groups (CRG) in 5′ 10× (grey) and SS2 (blue) datasets. **b** TCR beta-chain V- and J-gene usage by GLIPH CRG-1 clones. **c** Distribution of CRG-1 CD8 cells within each of the synovial T-cell clusters by number and as a percentage of the clusters. Top six clusters shown in each graph, statistics in Supplementary Data 7b. **d** Cells from CRG-1 highlighted on UMAP plot of all SFMC-derived T cells from 5′ 10× data set. **e, f** Volcano plots showing differential gene expression of synovial CD8 T cells in the HLA-DR-high cluster (**e**) and ZNF683+ cluster (**f**), against all other synovial CD8 T cells (Supplementary Data 3d, e, respectively). Two-sided Wilcoxon Rank Sum test with Bonferroni correction, labelling cut-offs for average logFC ± 0.65 (**e**), ±0.55 (**f**) and adjusted p value (≤1E−04) indicated by dotted lines. Source data are provided as a Source Data file.

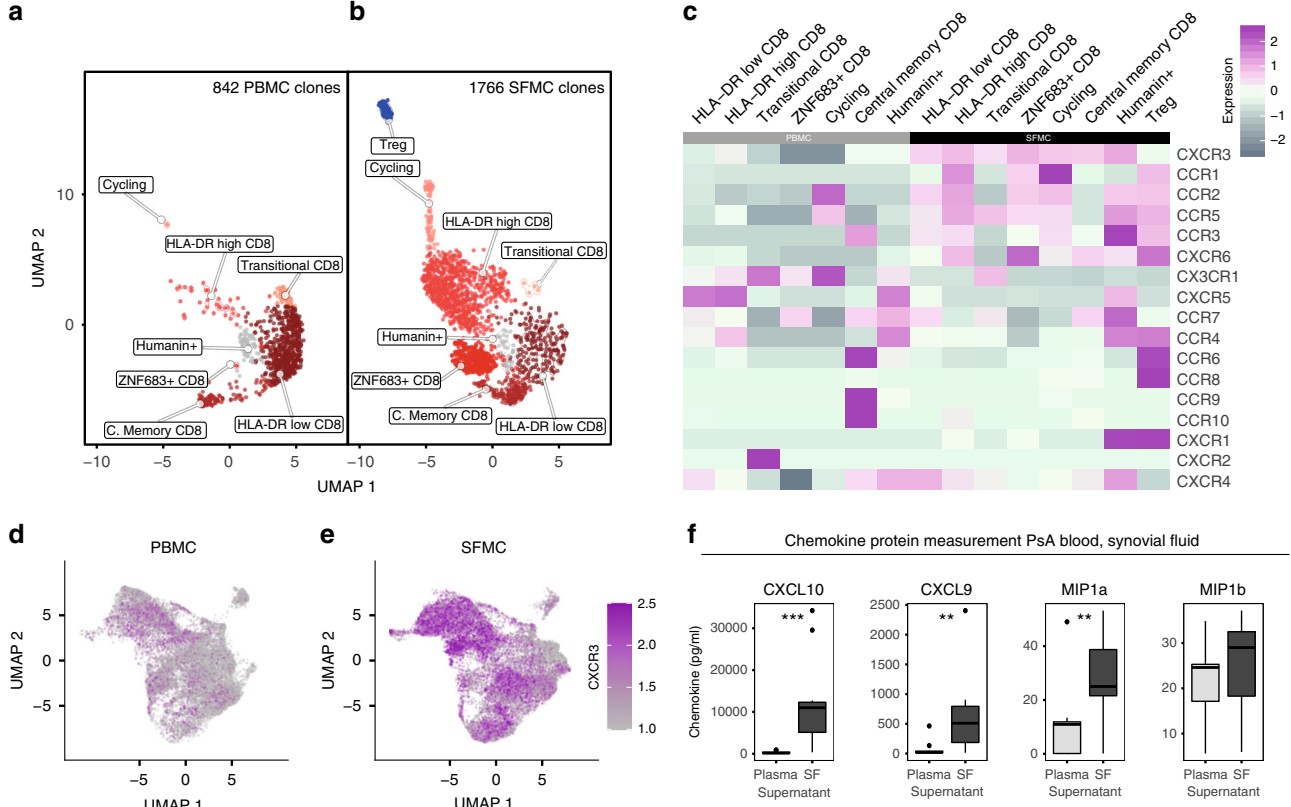

**Fig. 5 Clonal T-cell trafficking in psoriatic arthritis. a**, **b** UMAP of 2608 (1766 synovial, 842 peripheral blood) T cells from clones significantly enriched (adjusted $p$ value $\leq$ 0.05, two-sided Fisher's exact test with Benjamini–Hochberg correction for three patients) in PsA synovial fluid and blood, respectively, split by tissue of origin. Peripheral blood cells in panel **a**, synovial fluid cells in panel **b**. **c** Heatmap of significantly expanded blood and synovial T-cell clone clusters (columns) split by tissue of origin and showing chemokine receptor expression (rows). From paired blood and synovial fluid of three patients. **d** UMAP of 5′ 10× data set T cells derived from PBMC of three patients with *CXCR3* expression highlighted. **e** UMAP of 5′ 10× data set T cells derived from SFMC of three patients with *CXCR3* expression highlighted. **f** CXCL10, CXCL9, MIP1α and MIP1β protein quantification by LegendPlex in paired plasma and synovial fluid supernatant from 11 independent PsA patients, all replication attempts were successful. For all boxplots, the upper whisker extends to the largest value, maximally 1.5× IQR from the 75th centile, and the lower whisker extends to the smallest value, 1.5× IQR from the 25th centile. The junctions of whisker and box (hinge) represent the 25th and 75th centiles, while median is indicated by the central line. Outlier values are plotted individually beyond the whiskers. Two-sided paired $t$ test with Bonferroni correction (** = $p < 0.01$, ***= $p < 0.001$). Exact $p$ values CXCL10 = $8.5^{-5}$, CXCL9 = 0.0052 and MIP1a = 0.0055. Source data are provided as a Source Data file.

shared TCR recognition across patients provides the strongest evidence yet that PsA is an MHC-I peptide antigen complex-driven disease and argues against cytokine- or superantigen-driven expansion. In addition, we also note some CD4 clonal expansion predominantly in the Treg cluster (Fig. 3c, Fig. 5b, Supplementary Data 5), suggesting an antigen-driven expansion of this population in an attempt to regulate the inflammatory response.

An interesting question posed by this study is the role of tissue-resident memory T cells in the persistence of inflammation in the joints. Koebnerization (emergence at sites of trauma) of psoriatic skin lesions is well recognised[18], and large-joint oligo PsA often recurs in the same joint[19]. In addition to the description of ZNF683+ (tissue-resident) CD8 T cells in the skin[15], a recent report replicated the expansion of tissue-resident CD8 T cells in PsA synovial fluid and showed these cells to have a proinflammatory polyfunctional phenotype[20]. The origin of these ZNF683+ T cells is intriguing, and the existence of a single clone in several phenotypic cell states suggests bifurcation in T-cell fate at some point during proliferation. This phenomenon has previously been reported in response to infectious challenge[21].

Whilst the synovial tissue leucocyte (CD45+) compartment appears broadly similar to synovial fluid, being mainly composed

of CD3+ T cells and macrophages, subtle differences are observed. B cells and plasma cells, largely lacking in synovial fluid samples, account for a considerable proportion of synovial tissue cells, indicating a potentially differential ability of lymphocytes to traffic from the tissue to the fluid, or a selective survival advantage of B-cell lineages in the tissue microenvironment (for which there is significant evidence in RA[22]). The presence of B cells is particularly intriguing since the role of B cells in PsA disease pathogenesis is unknown with no clear autoantibodies described. Notably, ectopic lymphoid neogenesis has been described in the joints in PsA[23]. The majority of the synovial tissue CD8 cells associated with a HLA-DR-low cluster phenotype and expressed fewer activation markers compared to synovial fluid CD8 T cells. We cannot be sure if differences in disease state may account for this observation, and larger studies will be needed to understand these apparent differences in greater detail. Interestingly, HLA-DR- high and -low CD8 clusters were also reported in a recent single-cell study of synovial tissue CD8 cells in rheumatoid arthritis, highlighting potential parallels across these diseases[24]. One limitation of our study is that we were unable to obtain synovial tissue and fluid from the same individuals, which would have helped elucidate the relationship between tissue and fluid T cells.

Our data support a potentially important pathogenic role for the chemokine receptor CXCR3 in PsA pathogenesis. Increased CXCR3 gene expression in bulk PsA synovial fluid RNA preparations has been described previously, along with expression of one of its ligands. Here we show that *CXCR3* is highly expressed on synovial T cells (together with *CXCR6*, *CCR5* and *CCR2*), and that both its ligands CXCL9 and CXCL10 are also highly expressed in the (corresponding) synovial fluids. CXCR3 is known to be expressed on activated Th1 and CD8 T cells, and plays a key role in chemotaxis during inflammation[25, 26]. Interestingly, a reduced number of CXCR3-expressing T cells has previously been shown in the peripheral blood of patients with psoriasis, which had been speculated to be due to recruitment of these cells into skin lesions[27]. A similar mechanism may operate in PsA with preferential homing to inflamed joints. In summary, our findings raise the possibility that CXCR3+ CD8 T cells play a central role in executing localised inflammation in PsA, and thus represent an attractive therapeutic target.

A key question raised by this study is the nature of the antigen or antigens that we propose are driving the clonal expansion of CD8 T cells in PsA. LL37 has been proposed as an antigen recognised by CD8 T cells in psoriatic skin, but it will be intriguing in follow-up work to understand if common antigens are recognised in the skin and joint[28]. In particular, Cheuk and colleagues showed that *TRBV27* was significantly enriched in the CD49a+ epidermal T-cell compartment of all three donors sampled in their study[15]. We believe that the similarities of this independent study with ours are striking, especially as we also identify enrichment of TRBV27 in the ZNF683+ population, which also expresses *ITGA1* (CD49a). With application of novel approaches to identify antigens using MHC/peptide yeast-display libraries[29] or to potentially predict antigens directly from TCR sequences, these data may in the future allow us to define the nature of antigens that drive PsA.

## Methods

**Study participants**. Blood and synovial fluid samples were collected with written informed patient consent from PsA patients undergoing intra-articular aspiration at Oxford University Hospitals. The study was performed in accordance with protocols approved by the Oxford Research Ethics committee (Ethics reference number 06/Q1606/139). Synovial tissue was obtained from knee joints of PsA patients prior to commencement of any immunomodulatory therapy, including systemic corticosteroids by a minimally invasive, ultrasound-guided technique with written informed consent[30]. Protocols for these procedures were approved by relevant research ethics committees (reference number 12/NE/0251 Newcastle and West Midlands Black Country Research Ethics committee 07/H1203/57 (Birmingham)). Upon collection of biopsies, they were immediately cryopreserved in cryopreservation media Cryostor CS10 (Sigma—MERCK)[31]. Demographics for all patients enrolled in this study are listed in Supplementary Table 1.

**CyTOF staining and analysis**. Whole blood or synovial fluid were fixed with high-purity paraformaldehyde within 30 min of sample collection. Fixed blood was lysed with Permeabilization Buffer (eBioscience). Cells were stained in Maxpar staining buffer (Fluidigm) with antibodies listed in Supplementary Table 2. Samples were run on a Helios system alongside normalisation beads (Fluidigm). As samples were run fresh, each paired sample was analysed separately using a custom R work-flow[32], with cell populations clustered using the FlowSOM algorithm version 1.14.0[8].

**Cell isolation for flow cytometry cell-sorting**. SFMCs and PBMCs were freshly isolated within 30 min of sample collection by density-gradient centrifugation using Histopaque (Sigma). Cells were stained immediately and FACS-sorted for droplet-based single-cell RNA sequencing. Single-cell suspensions of freshly isolated paired SFMC and PBMC samples from three PsA patients were stained by a panel of fluorescently conjugated antibodies in staining buffer (RNAse-free PBS, 2 mM EDTA). The following antibodies were used, all from Biolegend: CD3-FITC (SK7, catalogue number 344803, dilution 1:50), CD4-APC (RPA-T4, catalogue number 300514, dilution 1:50), CD8a-PE (RPA-T8, catalogue number 301007, dilution 1:50) and CD45RA-BV421 (HI100, catalogue number 304129, dilution 1:50), together with eFluor780 viability dye (eBioscience, catalogue number 65-0865-14, dilution 1:250). Memory-enriched (CD45RA -negative) CD3+, CD4+,

CD8− and CD3+ CD8+ CD4− cells were sorted in a 1:1 ratio from both blood and synovial fluid of patients. Cryopreserved knee synovial tissue biopsies from two PsA patients were thawed and then digested in digestion buffer cocktail (50ug/ml Liberase$^{TM}$, Sigma and 40ug/ml DNAse I, Roche, in RPMI) in a 37 °C water bath for 30 min with continuous stirring with a magnetic stir bar in a U-bottom polystyrene tube ($12 \times 75$ mm$^2$). At 15 min during the digestion, samples were passed gently 10 times through a 14-G syringe needle for additional mechanical disruption. After digestion, the cells were filtered using a 70-um filter and washed prior to staining with antibodies. Following digestion, cells were counted by trypan blue to assess cell quantity and viability. Cell suspensions were stained with antibodies against CD45-APC (2D1, BD biosciences, catalogue number 340190, dilution 1:10), and the following antibodies from Biolegend: CD31-PE/Cy7 (WM59, catalogue number 303117, dilution 1:200), CD326-PE/Cy7 (9C4, catalogue number 324221, dilution 1:50), CD235a-PE/Cy7 (HI264, catalogue number 349111, dilution 1:20) and CD117-PE/Cy7 (104D2, catalogue number 313211, dilution 1:50). Cells were also stained with CD3-BUV395 (SK7, BD biosciences, catalogue number 564001, dilution 1:20) to establish post-sorted CD3+ fraction (Supplementary Fig. 3c). DAPI (Sigma) was added to the stained cells immediately prior to FACS sorting to isolate live leucocytes prior to loading onto separate channels of the 10× chromium machine.

**Droplet-based single-cell RNA sequencing**. For 3′ experiments, freshly isolated PBMC and SFMC were isolated using density centrifugation, washed and immediately loaded onto chromium controller (10×-Genomics) chip following the standard protocol for the Chromium single cell 3′ Kit (10× Genomics). For 5′ experiments, sorted memory-enriched CD4 and CD8 T-cell suspensions from peripheral blood and synovial fluid were prepared. Cells were counted and loaded into the chromium controller (10×-Genomics) chip following the standard protocol for the chromium single cell 5′ Kit (10× Genomics). The total time taken from sample retrieval to sample on the chip was 4 h. A cell concentration was used to obtain an expected number of captured cells between 5000 and 10,000 cells. All subsequent steps were done based on the standard manufacturer's protocol. Libraries were pooled and sequenced across multiple Illumina HiSeq 4000 lanes to obtain a read depth of approximately 30,000 reads per cell for gene-expression libraries and 8500 reads per cell for V(D)J-enriched T-cell libraries.

CD45+ synovial tissue cells were similarly processed using the 10× genomics chromium controller within 5 h of thawing, then sequenced using the NovaSeq sequencer and S2 flowcell to a read depth of over 1,000,000 reads per cell to achieve an estimated sequencing saturation of over 97% for both samples. Mean-used read pairs per cell for V(D)J libraries were over 35,000 for both samples.

**Plate-based single-cell RNA sequencing**. Freshly sorted CD45RA-negative CD3+CD4+, and CD3+CD8+ single cells from four patients were individually flow-sorted into 96-well full-skirted plates (Eppendorf) containing 10 μL of 2% Dithiothreitol (DTT, 2 M Sigma-Aldrich), RTL lysis buffer (Qiagen) solution. Cell lysates were sealed, mixed and spun down before storing at −80 °C. Paired-end multiplexed sequencing libraries were prepared following the Smart-seq 2 (SS2) protocol[33] using the Nextera XT DNA library prep kit (Illumina). A pool of bar-coded libraries from four different plates were sequenced across two lanes on the Illumina HiSeq 2500.

**Droplet-based single-cell RNA-seq mapping and pre-processing**. The raw single-cell sequencing data were mapped and quantified using the 10× Genomics Inc. software package CellRanger (v2.1 for peripheral blood and synovial fluid samples and v3.0.2 for synovial tissue samples) against the GRCh38-1.2.0 reference provided by 10×. Using the table of unique molecular identifiers produced by CellRanger, we identified droplets that contained cells using the call of functional droplets generated by CellRanger.

**Quality control of droplet-based single-cell data**. After cell-containing droplets were identified, gene-expression matrices of synovial fluid and peripheral blood-derived cells were first filtered to remove cells with >10% mitochondrial genes, <500 or >3500 genes and >25,000 UMI. Cells were further filtered to exclude multiplets defined as cells with greater than 1 beta chain or greater than 2 alpha chains where TCR sequencing data were available, and cells having both CD4 and CD8 gene expression where cells were sorted as either CD8+ or CD4+ by FACS. Matrices of synovial tissue-derived cells were similarly filtered to exclude cells with >10% mitochondrial genes, <200 or >2000 genes and >10,000 UMI. No multiplets were identified in synovial tissue sample data based on available TCR sequences.

**QC of SS2-based single-cell expression data**. Cells with more than 5 median absolute deviations (MAD) 8.35% of their mRNA originating from mitochondrial genes, a total number of reads <500,000 or >5,000,000, a total number of counts >3 MAD or with the number of genes <1000 or >6000, were filtered out prior to downstream analysis. Where matching TCR data for a cell was available, any cells with greater than 1 beta chain or greater than 2 alpha chains were additionally filtered.

**3′ analysis of peripheral blood and synovial fluid**. Sequencing reads from six datasets (three PBMC, three SFMC) were mapped with CellRanger V2.1.0 GRCh38 (ENSEMBL annotation). Quality control, filtering and clustering analysis was carried out with the R package Seurat version 3.0.0. To exclude low-quality cells, cells with fewer than 500 genes were excluded. Likely doublets were removed by filtering out cells with greater than 2500 genes or 10,000 UMIs. All cells with a mitochondrial fraction greater than 5% were also excluded. In total, 8606 PBMC cells and 7221 SFMC cells passed the filtering. The 6 datasets were SCT-transformed and then merged using the top 3000 variable genes. The RunPCA function was used to calculate 30 principal components, and SNN graph constructed using 30 dimensions and the SCT assay. Clustering was then performed on the SNN of the merged SCT assay at a resolution of 0.8 with otherwise default parameters, which yielded a total of 15 clusters classified by differentially expressed genes (Supplementary Fig. 2a) and visualised by UMAP.

**5′ integration of peripheral blood, synovial fluid and tissue**. Cellranger 5′ 10× output included eight expression matrices derived from the paired blood and synovial fluid T-cell-sorted (Supplementary Fig. 3) samples of three PsA patients, and synovial tissue leukocyte samples from two additional PsA patients. Downstream analyses of these matrices were carried out using R (version 3.6.1) primarily utilising the Seurat package (v3.1.4, satijalab.org/seurat).

CD45+ synovial tissue samples were first analysed as a whole. QC-filtered matrices from both patients were individually normalised using the SCTransform algorithm (min_cells = 1, return.only.var.genes = F) before merging into a single SCTransform-based (SCT) assay and determining the top 2000 variable genes, excluding TCR and BCR genes, of that assay. The Shared Nearest Neighbour (SNN) graph was then calculated using 30 dimensions, and 9 clusters were identified at a resolution of 1.3 according to the standard Seurat workflow (Supplementary Fig. 4). Cells belonging to T-cell clusters 2, 3 and 8 from this analysis and found to express at least 1 RNA transcript of CD3E were identified for downstream analysis as synovial tissue T cells.

To compare T cells across all sample types, QC-filtered blood and synovial fluid matrices were subsampled to include an equal number of cells for all patients (6867 cells per sample, totalling 41,202 cells) and analysed together with previously identified QC-filtered synovial tissue T cells of two patients. All eight datasets were individually normalised using the SCTransform function (min_cells = 1, return. only.var.genes = F). The eight datasets were then merged, and the top 2000 variable genes, excluding TCR and BCR genes, were identified from the SCT assay. The RunPCA function was used to calculate 30 principal components, and SNN graph constructed using 30 dimensions and the SCT assay. Clustering was then performed on the SNN of the merged SCT assay at a resolution of 0.65 with otherwise default parameters, which yielded a total of 16 clusters classified by differentially expressed genes (Supplementary Fig. 5) and visualised by UMAP (Fig. 2a).

For a focused analysis of clones enriched in either peripheral blood or synovial fluid, the six QC- filtered, SCTransformed datasets of T cells from both sample types were merged, and all cells corresponding to clones enriched in either sample type subsetted. Only cells belonging to the same sample type in which the clone was enriched were considered. The resulting clonally enriched subset consisted of 1766 synovial fluid and 842 peripheral blood T cells. The top 2000 variable genes excluding TCR genes were calculated from the SCT assay of this enriched subset, and 75 principal components calculated using the RunPCA function before constructing a SNN graph based on 75 dimensions and the SCT assay. Clustering was then performed on the SNN of the merged SCT assay at a resolution of 0.5 with otherwise default parameters, which yielded a total of eight clusters classified by differentially expressed genes (Supplementary Data 1e) and visualised by UMAP split across both sample types (Fig. 5a, b).

**Droplet- and SS2-based integration**. For validation of sequencing data across platforms, QC-filtered 10× expression matrices from one patient (PSA1607) were subsampled to include an equal number of T cells from blood and synovial fluid (12,200 cells from each, totalling 24,400 cells). QC-filtered SS2 data matrices from this same patient were similarly subsampled to include 433 T cells from both blood and synovial fluid. As all samples were from the same patient and of the same sorted cell type, an integration approach was adopted that utilised the larger 10× samples as a reference to project clustering onto the smaller number of SS2 cells.

In all, 10× and SS2 datasets were first individually normalised using the SCTransform function. Integration features, excluding TCR genes, were then calculated using the Seurat SelectIntegrationFeatures function and passed as an argument to the FindIntegrationAnchors function run with an SCT normalisation method and 10× datasets as reference. The Seurat IntegrateData function was finally used to integrate all sample datasets from the two single-cell sequencing platforms based again on the SCT normalisation method. The normal Seurat pipeline was then followed, calculating 50 principal components and constructing a SNN graph using 50 dimensions and the integrated assay. Clustering was performed on the integrated assay at a resolution of 0.65 with otherwise default

parameters, which yielded a total of 15 clusters classified by differentially expressed genes (Supplementary Data 1d) and visualised by UMAP (Supplementary Fig. 8).

**TCR mapping**. Chromium 10× V(D)J single-cell sequencing data were mapped and quantified using the software package CellRanger (v2.1 for peripheral blood and synovial fluid samples and v3.0.1 for synovial tissue samples) against the GRCh38 reference provided by 10× Genomics with that release. The generated consensus annotation files for each patient and sample type (blood or synovial fluid) were then used to construct clonality tables and input files for further downstream analysis. Full-length, paired TCR nucleotide sequences from SS2 data were constructed using the TraCeR program[34], and further mapped to V, D and J genes as well as CDR3 nucleotide and amino acid sequences using the online IMGT-HighVQuest tool (v3.4.13)[35]. The software package GLIPH v1.0 was used to construct and assess specificity groups[12]. As the GLIPH algorithm only makes use of single-CDR3 beta-chain amino acid sequences to associate clones of common specificity, multiple beta-chain sequences within the same partition were treated as multiplets and provided as separate individual sequences to GLIPH (annotated with a v suffix). Partitions containing only alpha-chain sequences were excluded from GLIPH input. The VDJtools 1.1.8[36] software package was used to construct circle plots illustrating V(D)J-gene usage.

To assess clonal enrichment, the proportion of cells having the same clone was compared between sample types for each clone using a two-sided Fisher's exact test with Benjamini and Hochberg (1995) correction for multiple comparisons (R Stats Package). A cell's clonotype was defined as the combined alpha- and beta-chain CDR3 nucleotide sequences for that cell. As it was not possible to deduce beta- and alpha-chain pairing for partitions with multiple beta chains, these partitions were treated as a single clone. When analysing both gene expression and clonality of the same cells, partitions containing more than one beta chain or more than two alpha chains were excluded from analysis as multiplets.

**Plate-based single-cell RNA-sequencing expression**. To assess the expression from the SS2 data, raw reads were pseudo-mapped and counted using kallisto v0.43.1[37] based on the annotation made by ENSEMBL(v90) of the human reference genome (GRCh38). To obtain per-gene counts, all of the transcript counts were summarised using scater v1.6.3[38].

**Chemokine protein quantification**. Paired plasma and synovial fluid supernatant was collected from patients undergoing knee aspiration procedures and frozen at −80 °C within 1 h of sample collections. Samples were thawed, and chemokine protein quantification was performed using a LEGENDplex™ Human Proinflammatory Chemokine Panel (13-plex) immunoassay (Cat# 740003) according to the manufacturer's instructions. Samples were acquired on two different dates, with similar results obtained. The data were acquired on a Novocyte instrument and analysed using the LEGENDplex™ Data Analysis Software (v8.0) provided by BioLegend.

**Data visualisation**. UMAP plots were generated using umap-learn (0.3.10) executed within a Python (3.6.10), numpy (version 1.18.1) environment.

**Statistical analyses**. Statistical tests were performed as indicated in the figure legends.

**Reporting summary**. Further information on research design is available in the Nature Research Reporting Summary linked to this article.

## Data availability
The datasets generated and/or analysed during the current study are available from the corresponding author on reasonable request. Sequencing data that support the findings of this study have been deposited in ArrayExpress with the accession code E-MTAB-9492 and in the European Genome-phenome Archive (EGA) with accession code EGAS00001002104. Source data are provided with this paper.

## Code availability
All code used to analyse data and synthesise the figures is available on request from the authors.

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

## Acknowledgements

We thank Jon Webber and Catherine Simpson for assistance with the cell sorting. We are grateful to Muzlifah Haniffa for critical review of the paper. The research was funded by The Kennedy Trust Studentship (F.P.), The Academy of Medical Sciences Grant to H.A. (SGL018\1006) and Personal fellowship support H.A. (Oxford-UCB Prize fellowship) S.B. (Wellcome, St. Baldrick's Foundation), C.B. (Wellcome). P.B. and H.A. are funded by the National Institute for Health Research (NIHR) Oxford Biomedical Research Centre (BRC); A.F. is funded by the NIHR Birmingham BRC at the University Hospitals Birmingham NHS Foundation Trust and the University of Birmingham. Work in the JDI laboratory is supported by the National Institute of Health Research (NIHR) Newcastle Biomedical Research Centre at Newcastle Hospitals Foundation Trust and Newcastle University and Versus Arthritis Research into Inflammatory Arthritis Centre, ref. 22072. The views expressed are those of the author(s) and not necessarily those of the NHS, the NIHR or the Department of Health.

## Author contributions

H.A.-M. and S.B. conceived and designed the experiments; F.P., L.N.N. and H.A.-M. performed the 10× experiments, designed and performed the computational analysis aided by M.D.C.V.H., M.D.Y., L.M., M.E. and D.S.; M.D.C.V.H. performed the SMART-seq 2 experiments assisted by C.G. and L.M.; N.Y. performed the CyTOF experiments assisted by A.L.L., S.C. and A.M.; N.Y., A.L.L. and S.C. performed the flow cytometry and assisted with the cell sorting; A.L.C. performed the protein quantification assays; A.G.P., K.R. and A.F. curated the pretreatment early arthritis cohorts and performed synovial biopsies; L.N.N. performed the synovial tissue processing; S.T., J.D.I. and C.B. contributed to the discussions; H.A.-M., S.B. and P.B. wrote the paper; P.B., S.B. and H.A.-M. co-directed this study. All authors read and approved the paper.

## Competing interests

The authors declare no competing interests.
