## [Peer Review File · Nature Communications]

Reviewers' comments:

Reviewer #1 (Rheumatoid arthritis, animal model, inflammation)(Remarks to the Author):

Psoriatic arthritis is an as yet poorly characterised pathogenesis and the application of advanced technologies such as those offered here are hoped to offer real insights. The authors present a very detailed informatic analysis of a relatively small cohort of patients.

Samples are derived from a group of patients from whom synovial fluid and blood can be coincidentally obtained – this large joint oligo group may, or may not, be representative of the wider PsA patient cohort and this a pragmatic approach yet implicitly brings in reservations about generalisation. Moreover, the analyses focus upon synovial fluid as opposed to synovial tissue - this is a significant limitation since it is in the latter compartment that most interest currently lies likely.

The cytof data are of interest but mainly provide an observational single time point (snap shot) of the contents of the synovial fluid versus blood compartments – this is a helpful insight but is not in itself particularly illuminating given what was known using bulk approaches previously (e.g. see Taams et al) – on the software and datasheets provided I was unable to read the actual list of antibodies used (supp Table 1 was corrupted) and as such I was uncertain how the actual cellular designates e.g. of a Th17 cell was made. The single cell analysis is performed with high fidelity as one would expect of this high quality centre and generated some fascinating data yet always derived from a rather small starting number of patient samples – this fundamentally limits the interpretative utility of the clusters identified and especially this is of concern when considering the T cell clonality. There are several prior papers, many quite old now, that have identified this kind of clonal asymmetry and as such a more extensive analysis would have been preferable. The risk is manifest in over interpretation e.g. in the comment that cluster 15 is of interest because it was found in all 3 donors and was proliferating – interesting yet, but conclusive, probably not in terms of strength of pathogenetic significance at this stage. The TCR sequencing is of interest and is perhaps the strongest part of this manuscript – it would have been helpful to have some kind of control here e.g. from RA fluids in which we have no evidence of CD8 clonality. As a minor issue, I was not quite clear from the description what the conclusions of CD4 expansions from this paper actually were.

Presence of clonotypes in fluid and blood potentially overstates the utility in targeting trafficking -this is hardly a novel comment given the extensive number of chemokine targeting studies performed over the years – one would seek much stronger evidence that these data to sustain further approaches of this nature. In this regard the protein estimations of chemokines in the final figures are somewhat anecdotal and do not really provide the functional correlates and insights one would have liked to see to really build a pathogenetic insight from this manuscript – measuring the protein presence is simply not providing functional validation.

A further minor issue – I was not easily able to understand the supplementary figure 4A/B – where do the synovial tissue data come from?

Reviewer #2 (Arthritis, autoimmunity, cytokine responses)(Remarks to the Author):

The claims of the paper identified in the abstract are interesting and novel for PsA. My major problem is that it is extremely difficult to follow. I think this is important because this was written for a multidisciplinary journal. Despite multiple re-reads I found it painful to understand how the data supported the interpretations made. I think that the data can be presented in a more reader friendly

manner. The presentation is very choppy. Some specifics that may help readers to better interpret the the conclusions of the paper:

- 1) lines 53-61 indicate in two places that Supp Figure 1 supports the interpretation. Which panels and why?
- 2) Line 71 refers to Supp Figure 2. There are 5 panels, a bit more explanation as to how these data explain how the cell clusters were defined. This figure is also mentioned on line 73.
- 3) In line 78 it clones are mentioned. How was a clone defined? Complete TCR sequence identify of multiple cells? Identical CDR3 sequences? What percent or number of identical cells is needed?
- 4) Line 84. There were 143 CGR with shared specificity? Can this number be statistically interrogated? We are referred to Supp Table 4. How shared is "shared specificity"?
- 5) line 95, Figure 2D-E. I'm not clear how to interpret the Figure. Actually the sentence beginning on line 93 is not clear to me as written.
- 6) is the increase of MKI67 alone enough to identify cells as actively proliferating? If they are actively proliferating, I would expect other related genes to be increased.
- 7) Line 77, Figure 2A-3. It would help to mention how to interpret the figures and what do normal control PB look like?
- 8) For the cell cluster annotations, please clarify the reference validating the strategies employed.
- 9) In general the going back and forth between the the different platforms is potentially a strength, however, the way it was presented was not easy to follow. Very choppy
- 11) Sentence beginning line 119, Supplemental Figure 3 is referred to. A little more explanation of how the data in the 6 panels supports the conclusion would be helpful.
- 10) Minor issues: It is hard to see the clusters in Figure 1f with the terms blocking the clusters. On line 70, tissue origin is referred to. Do the authors mean fluid?

Reviewer #3 (CyTOF, systems immunology)(Remarks to the Author):

The datasets described in this paper are extremely interesting and should be very useful to be published. From the mass cytometry analysis the authors conclude that there are differences in major cell subset frequencies such as: Naïve CD4, CD8, Memory CD4 and CD8, B cells, Basophils, pDC and cDC. This is not surprising and doesn't leverage the mass cytometry platform, which is ok. In the rest of the paper, much of the analysis is convoluted and not easy to follow. Without further justification, I do not think the Gliph clustering adds to the paper and unnecessarily confuses things. It is interesting that the authors have seen expanded clones in the synovial fluid, but this is not new. The trajectory analysis of one clone was performed but not clear what features of this would be consistent across other expanded clones. It is not clear to me if the authors are able to identify anything particular about the phenotypes of these clones when observed in the periphery. I think the analysis needs to be clarified before I can adequately comment on the validity of conclusions made in this paper.

Specific comments:

Please describe how mass cytometry samples were batched? Given that samples were fixed immediately after collection, it seems that multiple samples could be run in parallel to give some assessment of the variation seen for this perspective. Is the plot in Fig 1c,e from a single paired sample?

More specifics are needed to describe how FlowSOM clusters were merged between samples and quantified. Is tSNE used as part of the cluster definitions or just FlowSOM? There is mention in Fig. S2C of defining markers used for merging of clusters but this process is not described. Are these merged flowSOM cluster frequencies being plotted in Fig. 1d? Given the simplicity of findings being

reported for the mass cytometry section of this paper, it might be simpler to quantify frequencies using manual gating based on definitions provided in Fig. S2C. If this is done, please provide example gating.

The results section for Fig. 1F describes 4 patients but then figures describe only 3 patients. Please clarify.

Plots in Fig 2C are difficult to read. Please try log scale with smaller dots. Can these data be aggregated to make test the hypothesis of clonal enrichment in the synovial fluid?

I am unable to read Supplementary Table 3 and additional supplementary text is garbled in my version of the manuscript.

Please clarify statistics for determining enrichment of GLIPH TCR clusters between synovium and peripheral blood. Are there differences in the presence of any CRGs between these two compartments (n=3)? If not, please specify. I understand the logic of saying that a particular CRG contained the most expanded clones but I do not think this justifies the relevance or meaningfulness of this cluster. I also understand the concept of the GLIPH clusters but do not believe that there are only 143 antigen specificities being covered by the TCR sequences being analyzed here so I think there should be more skepticism in how this analysis is interpreted.

It is not possible for me to conclude that GRG1 were more predominantly assigned to clusters 4 and 10 from Figures G and H, which do not have any mention of cluster numbers. It is also not possible to conclude that these enrichments are not random because the total frequencies of cells within each of these clusters (Fig. 2H) are not provided. Please statistically test the null hypothesis that GRG1 cells have random phenotypes.

To say that new TCR sequences obtained using smart-seq 2 contain TCRs that are also assigned to CRG1 does not validate anything except the definition of CRG1. It would be more useful to test the hypothesis that the frequencies of cells within the CRG1 group differ in frequencies between tissue. If they aren't then I do not think it is worthwhile to further discuss the CRG1 group.

How was it concluded that clusters 4 and 10 were enriched for TRBV28 – line 112? (reference provided only about CD49a being a marker of tissue residence)

According to the plots shown in Fig. S4, cluster 4 seems to represent a large fraction of PBMC sample cells yet is being designated as "Activated". Could this observation of very high frequencies of activated (HLA-DR+) exhausted T cells in these three PBMC samples be validated by flow or mass cytometry? It would be good to determine what the cell surface protein expression profile of these non-cycling activated T cells are? Are these present at such high frequencies in healthy donors or are these high frequencies particular to these patients? Similar for cluster 3, do these patients really have such high frequencies of "exhausted cells" in blood and what does this phenotype really mean?

What is the difference in the UMAP imbedding for Fig. 2G vs. 2L? In 2G CRG1+ cells are all over the map, in 2L/2M, they are more restricted and fit into trajectories. What is different about how these UMAPs were run? I do not think it is fair to use CRG1+ cells are used in Fig. 2L/M. See above about why I question the utility of focusing on CRG1+ cells. Therefore, I do not find utility in the analysis shown for Fig. 2L/M. Instead trajectories should be described using the embedding from Fig. 2G. Analysis of individual clones such as in Fig. 2K is more meaningful, but not clear what can be

concluded from the analysis of this one clone. Is any aspect of this consistent across other clones identified.

I do not understand why new UMAP embeddings are needed for Fig. 3A, B? Are there any significant differences or skewing in the profiles of peripheral T cell clones (that overlap with clones in the synovium) as compared to the rest of the non-overlapping peripheral T cells? This analysis does not seem to address this question, which is implied by the title.

In Fig. 3D it is not clear why the PBMC and SFMC derived cells have identical looking UMAP plots – please clarify description of these plots.

Reviewers' comments:

Reviewer #1 (Rheumatoid arthritis, animal model, inflammation)(Remarks to the Author):

Psoriatic arthritis is an as yet poorly characterised pathogenesis and the application of advanced technologies such as those offered here are hoped to offer real insights. The authors present a very detailed informatic analysis of a relatively small cohort of patients.

Samples are derived from a group of patients from whom synovial fluid and blood can be coincidentally obtained - this large joint oligo group may, or may not, be representative of the wider PsA patient cohort and this a pragmatic approach yet implicitly brings in reservations about generalisation. Moreover, the analyses focus upon synovial fluid as opposed to synovial tissue - this is a significant limitation since it is in the latter compartment that most interest currently lies likely.

We thank the reviewer for these comments and insights. We specifically chose to look at the cellular landscape of PsA, and large joint involvement is a well recognised feature of this disease (Moll & Wright, Semin. Arthritis Rheum. 1973). We feel this remains a population with a significant amount of unmet clinical need and thus any additional insight into the pathobiology is of high interest to the field. Additionally, we show-case using these samples the power of single-cell paired gene and TCR expression and are able to gain valuable insights into human T cell biology directly ex-vivo in human chronic inflammation.

Based on the reviewers comments we have now included synovial tissue samples from the knees of PsA patients (now shown in Figure 2e/Supplementary Figure 4). A key focus of this paper is the analysis of the clonal architecture at single cell resolution. Due to the size limitations of tissue biopsy, it would be technically very challenging to get sufficient cell numbers required to statistically test for clonal expansion using those samples and therefore the use of synovial fluid is necessary here.

The cytof data are of interest but mainly provide an observational single time point (snap shot) of the contents of the synovial fluid versus blood compartments - this is a helpful insight but is not in itself particularly illuminating given what was known using bulk approaches previously (e.g. see Taams et al) - on the software and datasheets provided I was unable to read the actual list of antibodies used (supp Table 1 was corrupted) and as such I was uncertain how the actual cellular designates e.g. of a Th17 cell was made.

We apologise for the issue with opening the supplementary data files. This has now been addressed. Supplementary Figure 1 b and c shows markers used to define cell clusters by CyTOF. We believe this data advances the field because the clustering algorithms used were unsupervised; this has not previously been done in PsA synovial fluid analysis from other groups.

The single cell analysis is performed with high fidelity as one would expect of this high quality centre and generated some fascinating data yet always derived from a rather small

starting number of patient samples – this fundamentally limits the interpretative utility of the clusters identified and especially this is of concern when considering the T cell clonality. There are several prior papers, many quite old now, that have identified this kind of clonal asymmetry and as such a more extensive analysis would have been preferable. The risk is manifest in over interpretation e.g. in the comment that cluster 15 is of interest because it was found in all 3 donors and was proliferating – interesting yet, but conclusive, probably not in terms of strength of pathogenetic significance at this stage.

We recognise and discuss in our manuscript that clonal expansion in PsA has been demonstrated by a number of groups but this has always been carried out using bulk sequencing approaches. Our study allows us to understand the T cell receptor alpha-beta pairing and follow individual clones and further allows us to look at the gene expression signatures of those clones. This has not been done previously in PsA. We thus believe that the unprecedented number of T cells examined at the single cells level in our study in significant part mitigates the relatively small number of patients studied.

The TCR sequencing is of interest and is perhaps the strongest part of this manuscript – it would have been helpful to have some kind of control here e.g. from RA fluids in which we have no evidence of CD8 clonality. As a minor issue, I was not quite clear from the description what the conclusions of CD4 expansions from this paper actually were.

We thank the reviewer for this comment but due to limitation in resource we chose to focus this study on PsA. RA single cell sequencing has been recently done and published by the AMP group (Zhang et al, Nature Medicine 2019) and given their approach is based on tissue, and the inherent inability to assess clonal expansion using biopsy samples we feel an RA control group falls outside the scope of this manuscript. Because we only observed minor CD4 clonal expansions, we felt it was difficult to draw conclusions from those with any degree of confidence. The full data-set including the clonal enrichment of CD4 cells is included. Figure 3c and Supplementary Table 6 show that the majority of CD4 T cell clonal expansion was in Treg cells and this has now been added to the discussion (line 207-209).

Presence of clonotypes in fluid and blood potentially overstates the utility in targeting trafficking -this is hardly a novel comment given the extensive number of chemokine targeting studies performed over the years – one would seek much stronger evidence that these data to sustain further approaches of this nature. In this regard the protein estimations of chemokines in the final figures are somewhat anecdotal and do not really provide the functional correlates and insights one would have liked to see to really build a pathogenetic insight from this manuscript – measuring the protein presence is simply not providing functional validation.

We thank the reviewer for these comments. We agree that measuring chemokine concentrations does not provide strong functional validation but given the recent publications around CXCR3 in PsA (Muntyanu et al, Arthritis Research and Therapy 2016; Abji et al, Arthritis Rheumatol 2019), we feel that our findings are of relevance to the field. Full functional validation would require a clinical trial. We note that MDX-1100, a monoclonal antibody against CXCL-10 failed in clinical trials of rheumatoid arthritis, our data

highlights the over-expression of multiple CXCR3 ligands in the joints of PsA patients and would therefore imply this redundancy needs to be considered in modulating this axis therapeutically.

A further minor issue - I was not easily able to understand the supplementary figure 4A/B - where do the synovial tissue data come from?

Apologies, we have now corrected and clarified this. There was no synovial tissue in the original manuscript and this has now been added to the revised manuscript.

Reviewer #2 (Arthritis, autoimmunity, cytokine responses)(Remarks to the Author):

The claims of the paper identified in the abstract are interesting and novel for PsA. My major problem is that it is extremely difficult to follow. I think this is important because this was written for a multidisciplinary journal. Despite multiple re-reads I found it painful to understand how the data supported the interpretations made. I think that the data can be presented in a more reader friendly manner. The presentation is very choppy. Some specifics that may help readers to better interpret the the conclusions of the paper:

We apologise, the manuscript has been fully re-written and re-structured and we hope this makes it easier to follow and read.

1) lines 53-61 indicate in two places that Supp Figure 1 supports the interpretation. Which panels and why?

We have re-structured the supplementary figures and references to this data is clarified in the revised manuscript.

2) Line 71 refers to Supp Figure 2. There are 5 panels, a bit more explanation as to how these data explain how the cell clusters were defined. This figure is also mentioned on line 73.

As above

3) In line 78 it clones are mentioned. How was a clone defined? Complete TCR sequence identify of multiple cells? Identical CDR3 sequences? What percent or number of identical cells is needed?

Individual clones are defined as having the same alpha and beta chain CDR3 nucleotide sequences (Methods). Clones may consist of 1 or more cells. When analysing clonality and gene expression of cells, careful consideration was given to a strategy which would maximise the use of available datasets and minimise misinterpretation. During single cell sequencing, especially on the 10x platform, there is an expectation from the mechanical nature of separating cells that a number of "partitions" to which sequencing data is

attributed in reality represent multiple cells. It is desirable to exclude such partitions during analysis of gene expression (GEX) as these partitions may interfere with clustering and falsely associate the expression or magnitude of expression of genes with what is classified as a single cell. For this reason when analysing the cross-over between GEX and clonality, where more than 1 beta chain were present within a partition, such partitions were excluded.

When analysing overall clonality and the enrichment of clones present in either synovial fluid or peripheral blood, automatically excluding partitions with more alpha or beta chains than would be expected for a single cell would bias the proportion of cells represented by remaining clones upwards. For this reason, and to avoid overlooking any potential disease specific peculiarity in regards to clonality, clonal enrichment analysis did not exclude clones with multiple beta chains and more than 2 alpha chains from analysis. Cells belonging to such clones however maintained the same definition of a clone i.e. having the same alpha and beta chain CDR3 sequences. This approach is also adopted by 10x analysis tools such as the Vloupe browser, therefore additionally providing a cross reference to support the correct annotation of clones during R analysis.

The GLIPH algorithm does not take into consideration alpha chain CDR3 sequences when clustering. Therefore, to maximise the utility of GLIPH, all beta chain CDR3 sequences, even those found within the same partition, were passed as input to GLIPH.

4) Line 84. There were 143 CGR with shared specificity? Can this number be statistically interrogated? We are referred to Supp Tabe 4. How shared is “shared specificity” ?

Convergence groups (CRG) represent a collection of CDR3 beta sequences which are predicted by the GLIPH algorithm to be specific against a common antigen. The number of 143 CRG is not in itself considered of importance. Rather the reduction of this number to only 5 convergence groups common to a greater number of patients (6 patients in total) suggests recognition of specific disease relevant antigens. We further suggest that 1 of these 5 convergence groups in particular is most likely to be relevant to disease pathogenesis given the inclusion of CDR3 sequences strongly enriched in synovial fluid, which map onto cells with gene expression signatures suggestive of activation and proliferation. Regarding the degree of shared specificity that can be confidently attributed to a convergence group, the GLIPH algorithm has previously demonstrated the capacity to predict de novo sequences with shared specificity against an antigen in prior publications (Glanville, Jacob, Huang Huang, Allison Nau, Olivia Hatton, Lisa E. Wagar, Florian Rubelt, Xuhuai Ji, et al. ‘Identifying Specificity Groups in the T Cell Receptor Repertoire’. *Nature* 547, no. 7661 (21 June 2017): 94–98. <https://doi.org/10.1038/nature22976>.), however it is beyond the scope of this paper to further validate GLIPH.

5) line 95, Figure 2D-E. I’ m not clear how to interpret the Figure. Actually the sentence beginning on line 93 is not clear to me as written.

The manuscript has been re-written and the reference to the figure now clarified

6) is the increase of MKI67 alone enough to identify cells as actively proliferating? If they are actively proliferating, I would expect other related genes to be increased.

In addition to MKI67, we have also identified a strong association of STMN1, known to play a critically important role in regulating the cell cycle (Rubin, Camelia Iancu, and George F. Atweh. 'The Role of Stathmin in the Regulation of the Cell Cycle'. *Journal of Cellular Biochemistry* 93, no. 2 (2004): 242–50. <https://doi.org/10.1002/jcb.20187>), with the "Cycling" cell cluster, along with additional genes one might expect to be increased during proliferation, such as tubulin genes (TUBB, TUBA1A), and the Centromere Protein F gene (CENPF) (Supplementary Figure 5).

7) Line 77, Figure 2A-3. It would help to mention how to interpret the figures and what do normal control PB look like?

Previous Figures 2a-b now correspond to Figures 3a-b. Figure legends have been updated to describe these figures in more detail. The figures are designed to give a general overview of differences between V and J gene usage between peripheral blood and synovial fluid within our dataset, and for comparison also with our CRG-1 convergence group (Now Supplementary Figure 9b). Control blood data is not included as part of this study, however, as requested for comparison, V and J usage of 7 age and sex matched (+/- 6 years) healthy control samples (giving equal weighting to each sample) is provided below (data used to compute plot from Britanova, et al. 'Age-Related Decrease in TCR Repertoire Diversity Measured with Deep and Normalized Sequence Profiling'. *The Journal of Immunology* 192, no. 6 (15 March 2014): 2689–98. <https://doi.org/10.4049/jimmunol.1302064>.)

8) For the cell cluster annotations, please clarify the reference validating the strategies employed.

We have annotated cell clusters according to genes overexpressed within each cluster relative to cells from other clusters. We do not claim to identify novel cell types within this manuscript, recognising that clusters may represent phenotypic states of the same cell type. Supplementary Table 2 lists all differential gene expression analysis used to classify any clusters identified.

9) In general the going back and forth between the the different platforms is potentially a strength, however, the way it was presented was not easy to follow. Very choppy

The manuscript has now been re-written for better flow.

11) Sentence beginning line 119, Supplemental Figure 3 is referred to. A little more explanation of how the data in the 6 panels supports the conclusion would be helpful.

This section has been re-written with a focus on clonal phenotype.

10) Minor issues: It is hard to see the clusters in Figure 1f with the terms blocking the clusters. On line 70, tissue origin is referred to. Do the authors mean fluid?

We apologise for the confusion. This figure has been redone and the words "sample type" used in place of tissue.

Reviewer #3 (CyTOF, systems immunology)(Remarks to the Author):

The datasets described in this paper are extremely interesting and should be very useful to be published.

We thank the reviewer for recognising the value of our manuscript.

From the mass cytometry analysis the authors conclude that there are differences in major cell subset frequencies such as: Naïve CD4, CD8, Memory CD4 and CD8, B cells, Basophils, pDC and cDC. This is not surprising and doesn't leverage the mass cytometry platform, which is ok. In the rest of the paper, much of the analysis is convoluted and not easy to follow. Without further justification, I do not think the Gliph clustering adds to the paper and unnecessarily confuses things.

We thank the reviewer for these comments. We agree that the use of GLIPH makes certain assumptions on the data. We still think this approach is interesting but have now moved all GLIPH analysis to supplementary figures and tables and instead focus our conclusions on the individual clones. We do however note that GLIPH has been recently used in a publication to understand antigen recognition of clonally expanded CD8 cells in Alzheimer's disease (Gate et al, Nature 2020).

It is interesting that the authors have seen expanded clones in the synovial fluid, but this is not new. The trajectory analysis of one clone was performed but not clear what features of this would be consistent across other expanded clones.

We thank the reviewer for these comments. The trajectory analysis on smaller clones is difficult to do given the diminishing statistical power. Thus we have re-focused the discussion around individual clones existing in multiple cell states which is more robust and indeed true of almost all expanded synovial clones in our data-set (New Figure 3g and Supplementary Figure 7).

It is not clear to me if the authors are able to identify anything particular about the phenotypes of these clones when observed in the periphery. I think the analysis needs to be clarified before I can adequately comment on the validity of conclusions made in this paper.

Data regarding comparative clonal phenotypes in synovial fluid and blood is now included in a new heatmap in Supplementary Figure 7 and briefly commented on in the manuscript line 121-124.

Specific comments:

Please describe how mass cytometry samples were batched? Given that samples were fixed immediately after collection, it seems that multiple samples could be run in parallel to give some assessment of the variation seen for this perspective. Is the plot in Fig 1c,e from a single paired sample?

The mass cytometry samples were batched by matching blood and synovial fluid from the same patient with both fixed within one hour of patient recruitment, stained and run on the Helios at the same time therefore different patients could not be barcoded and batched together. We recognise the limitations of this approach and so only paired samples were clustered in the same analysis cycle. The plots in Fig 1c and e are indeed from a single paired sample – patient PsA1505. We have now clarified this in the figure legend.

More specifics are needed to describe how FlowSOM clusters were merged between samples and quantified. Is tSNE used as part of the cluster definitions or just FlowSOM? There is mention in Fig. S2C of defining markers used for merging of clusters but this process is not described. Are these merged flowSOM cluster frequencies being plotted in Fig. 1d? Given the simplicity of findings being reported for the mass cytometry section of this paper, it might be simpler to quantify frequencies using manual gating based on definitions provided in Fig. S2C. If this is done, please provide example gating.

tSNE is only used for visualisation of the clusters. The clustering was performed using the FlowSOM algorithm as described in reference 20. Each paired sample underwent a separate clustering analysis due to the lack of batching of patient samples. Therefore for each paired sample, there would be a heat map generated as in Supplementary Figure 1b, and using the defining markers listed in Supplementary Figure 1c, the clusters could be annotated and merged if necessary into broad cell populations. Merging of clusters occurred only if the cell populations had been over-clustered. We were able to quantitate the percentage of each cell population per paired sample following the clustering, and these figures were extracted and used to plot Fig 1d. We have not done manual gating with the mass cytometry data as we did not want the manuscript to be biased in terms of gating. By using unsupervised clustering analysis, the only introduction of bias is when annotating clusters and we have been fully transparent in our definitions. We chose to use FlowSOM as our preferred clustering method as described in the Weber & Robinson 2016 paper (*Cytometry A*. 89, 1084-1096).

The results section for Fig. 1F describes 4 patients but then figures describe only 3 patients. Please clarify.

We apologise for the confusion. Figure 1F no longer exists in the new version, instead Figure 2a amalgamates T cell data from 5 PsA patients (3 synovial fluid / blood and 2 synovial tissue)

Plots in Fig 2C are difficult to read. Please try log scale with smaller dots. Can these data be aggregated to make test the hypothesis of clonal enrichment in the synovial fluid?

We thank the reviewer for these comments. The main source of the cluttered appearance in these graphs stems from a high number of clones represented by only a handful of cells thus with low proportion of cells in both peripheral blood and synovial fluid. We have now modified the graph to remove clones with less than 10 cells in either blood or synovial fluid and have made points slightly smaller. Clutter has been reduced and the main purpose of these graphs in highlighting dominant enriched clones has been preserved. This change does not affect the display of any clones found to be significantly enriched in blood or synovial fluid of any patient (red points). Aggregating data across patients unfortunately re-introduces clutter and as many clones are exclusively present in either blood or synovial fluid, displaying on a log scale poses difficulties in plotting a zero value on either axis.

I am unable to read Supplementary Table 3 and additional supplementary text is garbled in my version of the manuscript.

We apologise that the files were corrupted. The supplementary data have all been comprehensively re-vised in line with the fully revised manuscript.

Please clarify statistics for determining enrichment of GLIPH TCR clusters between synovium and peripheral blood. Are there differences in the presence of any CRGs between these two compartments (n=3)? If not, please specify. I understand the logic of saying that a particular CRG contained the most expanded clones but I do not think this justifies the relevance or meaningfulness of this cluster. I also understand the concept of the GLIPH clusters but do not believe that there are only 143 antigen specificities being covered by the TCR sequences being analyzed here so I think there should be more skepticism in how this analysis is interpreted.

GLIPH is no longer the main focus of analysis and we have moved GLIPH associated figures to supplementary data, along with a re-write focussing instead on clonal enrichment in either blood or synovial fluid. An odds ratio and Fisher's exact test is now supplied to support the association of cells enriched in synovial fluid or from CRG-1 belonging to particular clusters (new Supplementary Table 7)

The number of 143 antigen specificities (convergence groups) pertains only to antigen specificities which were found to be shared across all 3 patients with available 10x V(D)J sequencing. Many more convergence groups were discovered for individual patients, or shared between only 2 patients (980 in total, Supplementary Table 8d).

It is not possible for me to conclude that GRG1 were more predominantly assigned to clusters 4 and 10 from Figures G and H, which do not have any mention of cluster numbers. It is also not possible to conclude that these enrichments are not random because the total frequencies of cells within each of these clusters (Fig. 2H) are not provided. Please statistically test the null hypothesis that GRG1 cells have random phenotypes.

GLIPH analysis has been moved to Supplementary figure 9, where an additional graph (9c) also indicates the proportion of a cluster represented by CRG-1 cells, with odds ratios specified in the figure legend and detailed statistical calculations provided in Supplementary Table 7.

To say that new TCR sequences obtained using smart-seq 2 contain TCRs that are also assigned to CRG1 does not validate anything except the definition of CRG1. It would be more useful to test the hypothesis that the frequencies of cells within the CRG1 group differ in frequencies between tissue. If they aren't then I do not think it is worthwhile to further discuss the CRG1 group.

We thank the reviewer for this valid comment. The manuscript has now been re-written, removing reference to validation and the GLIPH analysis has been moved to the supplementary part of the manuscript. In any case, to address this comment further we include a figure below that shows the proportion of synovial fluid cells represented by CRG-1 is vastly greater than in peripheral blood, where red dots represent CRG-1 cells.

How was it concluded that clusters 4 and 10 were enriched for TRBV28 - line 112? (reference provided only about CD49a being a marker of tissue residence)

In the Cheuk et al paper (Immunity 2017), Figure 2 panel F shows that *TRBV27* is significantly enriched in the CD49a+ epidermal T cell compartment of all three donors sampled in that study. In addition, *TRBV28* is also shown to be enriched in the epidermal CD49a+ compartment of one of the three donors. We believe the similarities of this independent study with ours is striking, especially as we also identify enrichment of *TRBV27* in the ZNF683+ population which also expresses *ITGA1* (CD49a). This has now clarified in the discussion (lines 243-247)

According to the plots shown in Fig. S4, cluster 4 seems to represent a large fraction of PBMC sample cells yet is being designated as “Activated” . Could this observation of

very high frequencies of activated (HLA-DR+) exhausted T cells in these three PBMC samples be validated by flow or mass cytometry?

This is an artefact of integrating data from blood and synovial fluid together. In the new analysis we firstly refrain from using terms such as activated and exhausted to define clusters and instead use specific markers (HLA-DR) without making further assumptions. We also adopt a less forceful approach to integration of data matrices, using the "merge" function (updated Methods) and making no assumptions regarding what cell types might be expected when integrating peripheral blood, synovial fluid and synovial tissue matrices. In addition we perform an analysis of blood vs synovial fluid for some of the key CD8 clusters and show that activation markers are enriched in the synovial fluid. Fig 2f-h

It would be good to determine what the cell surface protein expression profile of these non-cycling activated T cells are? Are these present at such high frequencies in healthy donors or are these high frequencies particular to these patients? Similar for cluster 3, do these patients really have such high frequencies of "exhausted cells" in blood and what does this phenotype really mean?

As above

What is the difference in the UMAP imbedding for Fig. 2G vs. 2L? In 2G CRG1+ cells are all over the map, in 2L/2M, they are more restricted and fit into trajectories. What is different about how these UMAPs were run? I do not think it is fair to use CRG1+ cells are used in Fig. 2L/M. See above about why I question the utility of focusing on CRG1+ cells. Therefore, I do not find utility in the analysis shown for Fig. 2L/M. Instead trajectories should be described using the embedding from Fig. 2G. Analysis of individual clones such as in Fig. 2K is more meaningful, but not clear what can be concluded from the analysis of this one clone. Is any aspect of this consistent across other clones identified.

Many thanks for your comments. We take these on board and now analyse the cell states of all expanded clones in Figure 3g and Supplementary Figure 7, and have removed the trajectory analysis.

I do not understand why new UMAP embeddings are needed for Fig. 3A, B? Are there any significant differences or skewing in the profiles of peripheral T cell clones (that overlap with clones in the synovium) as compared to the rest of the non-overlapping peripheral T cells?

This analysis does not seem to address this question, which is implied by the title.

We believe that the two datasets complement one another. By comparing cells from clones enriched in synovial fluid to those enriched in blood we focus specifically on 2 subsets of cells known to be disproportionately overrepresented in each of these 2 sample types. Mapping findings back to the larger dataset shows the degree to which conclusions drawn can be generalised. We now also include a differential gene expression analysis between SFMC and PBMC derived T cells from the larger integrated dataset for comparison (Supplementary Table 2g), where CXCR3 and CXCR6 are the only chemokine receptors found to be overexpressed in synovial fluid T cells (using the same average logFC cutoff of 0.25).

We attempted to distinguish the phenotype of cells with shared clonality present in both blood and SF, however limited cell numbers from the unenriched sample type made it difficult to draw meaningful conclusions.

In Fig. 3D it is not clear why the PBMC and SFMC derived cells have identical looking UMAP plots - please clarify description of these plots.

Further comparing PBMC and SFMC derived cells within clusters highlights key differences. This is now made more clear (Fig 2f-h), and as discussed in answering a previous question an updated integration approach has been adopted which is outlined in the revised methods section. The new integration approach which merges individually SCTransformed datasets makes it more likely for PBMC and SFMC cells to cluster separately if these cells represent fundamentally different cell types, while preserving distinctions between PBMC and SFMC derived T cells within the clusters.

REVIEWERS' COMMENTS:

Reviewer #1 (Remarks to the Author):

Thanks for addressing the major concerns previously raised. The rewritten manuscript is improved in flow. The addition of the synovial tissues is welcome but one is still left with a very small sample size which limits the generalisability of these data for this reviewer., not least given the heterogeneity of the PsA syndrome. The authors should not obfuscate in terms of the number of clones as opposed to the number of patients evaluated - the latter remains insufficient. This also pertains to the arguments around chemokine contributions which are not really addressed in the revised manuscript - this work remains highly speculative and observational and does not really test their hypothesis - it appears somewhat anecdotal which is unfortunate given the elegant sequencing biology that goes before.

Reviewer #2 (Remarks to the Author):

I forgot to ask in my original review, what was the total number of SFs from patients with PsA, what medications were they on, and was there any influence of medications on any of the results?

Reviewer #3 (Remarks to the Author):

My comments have all been adequately addressed. I think that the revised paper is nicely improved and much easier to follow.

Reviewer #1 (Remarks to the Author):

Thanks for addressing the major concerns previously raised. The rewritten manuscript is improved in flow. The addition of the synovial tissues is welcome but one is still left with a very small sample size which limits the generalisability of these data for this reviewer., not least given the heterogeneity of the PsA syndrome. The authors should not obfuscate in terms of the number of clones as opposed to the number of patients evaluated - the latter remains insufficient. This also pertains to the arguments around chemokine contributions which are not really addressed in the revised manuscript - this work remains highly speculative and observational and does not really test their hypothesis - it appears somewhat anecdotal which is unfortunate given the elegant sequencing biology that goes before.

We thank the reviewer for their comments and we are happy that they find the manuscript has been improved. We agree that limitation of this study is the small number of patients studied but we believe the methodology employed in this in-depth study will advance the field by providing a template for how these studies can be carried out across a large number of patients.

Reviewer #2 (Remarks to the Author):

I forgot to ask in my original review, what was the total number of SFs from patients with PsA, what medications were they on, and was there any influence of medications on any of the results?

None of the PsA patients who donated samples to the sequencing experiments were on active DMARD or biologic medication at the time of enrolment. The patient demographics are now included in supplementary table 1. We specifically enrolled patients who were not taking active therapy so that we can study the disease state rather than the influence of medications. We believed this would be the best starting point given the relatively small sample size for the sequencing experiments.

Reviewer #3 (Remarks to the Author):

My comments have all been adequately addressed. I think that the revised paper is nicely improved and much easier to follow.

We thank the reviewer for their insightful comments in the first review which greatly helped us improve this manuscript.